# Unroofed Coronary Sinus in a Dog: Diagnostic Utility of ECG-Gated Computed Tomography

**DOI:** 10.3390/ani15192834

**Published:** 2025-09-28

**Authors:** Nanaha Ito, Risa Okamoto, Kazumi Shimada, Daigo Azakami, Zeki Yilmaz, Ryou Tanaka, Lina Hamabe

**Affiliations:** 1Laboratory of Veterinary Clinical Oncology, Faculty of Veterinary Medicine, Tokyo University of Agriculture and Technology, Tokyo 183-8509, Japan; 2Animal Medical Center, Tokyo University of Agriculture and Technology, Tokyo 183-8509, Japan; 3Laboratory of Veterinary Surgery, Faculty of Veterinary Medicine, Tokyo University of Agriculture and Technology, Tokyo 183-8509, Japan; 4Department of Internal Medicine, Faculty of Veterinary Medicine, Uludag University, Bursa 16059, Turkey

**Keywords:** congenital heart defect, dog, ECG-gated computed tomography, unroofed coronary sinus syndrome

## Abstract

Due to technological improvements in echocardiography, most cardiac problems can be diagnosed using transthoracic echocardiography. However, when there is a substantial change in the cardiac morphology, it may be insufficient for diagnosis. Electrocardiogram (ECG)-gated computed tomography (CT) is a type of exam that synchronizes with the patient’s ECG to capture a detailed image of the diastolic and systolic cardiac morphology, with less cardiac-motion-induced artifacts, allowing for a better insight and a more definitive diagnosis. This case report describes a rare instance of an unroofed coronary sinus syndrome, which was challenging to diagnose by conventional echocardiography but was successfully diagnosed using ECG-gated CT scans.

## 1. Introduction

Unroofed coronary sinus syndrome (UCSS) is a rare congenital heart disease similar to atrial septal defect (ASD). Both are conditions in which the left atrium (LA) and the right atrium (RA) become connected, and an unusual blood flow from the LA to the RA forms [1]. In ASD, the connection is formed through an opening in the atrial septum, while in UCSS, it is formed through an opening in the coronary sinus (CS) that connects to the LA [2]. The CS is a vein that collects blood from the cardiac muscle and transports it to the RA. In UCSS, the irregular flow of blood from the LA to the CS causes congestion in both the CS and the RA [3]. There are two ways a connection between the CS and the LA could form. The CS and the LA may connect indirectly, forming a vein between the CS and the LA, or they may directly connect through an opening between the lateral wall of the CS and the LA cavity [4].

As UCSS is an extremely rare condition, the little-known information comes from previous reports of humans. The information on actual conditions that follow an animal, particularly a dog’s UCSS, remains sparse since only 3 cases of dogs with it have been reported [5,6,7]. In many human cases, UCSS can be observed in conjunction with a persistent left superior vena cava (LSVC), which may also include a connection between the two superior venae cavae [3]. In humans, common clinical symptoms such as cyanosis, dyspnea, palpitations, cardiac murmur, and arrhythmias can be seen [8].

Transthoracic echocardiography (TTE) is the standard examination when diagnosing any kind of cardiac problem. Although diagnosing heart disease has become possible with advancing technology in echocardiography, this is not always the case. When diagnosing atypical congenital heart diseases, 2D echocardiography may be unable to depict the inside of the heart due to the substantial changes in its shape, leading to an uncertain diagnosis. In such cases, contrast echocardiography- including a bubble study performed via left brachial vein- may offer an advantage in the preliminary diagnosis and/or confirmation of UCSS [9].

Electrocardiogram (ECG)-gated computed tomography (CT) synchronizes with ECG and captures an image of one heartbeat, depicting diastole and systole cardiac morphology. Modern 64-slice CT scanners can obtain highly detailed images of the thoracic aorta and other cardiac structures due to their short acquisition time and high resolution in both temporal and spatial fields [10]. Additionally, ECG-synchronized reconstruction allows an accurate presentation of the angiocardiac system with few cardiac-motion-induced artifacts, enabling us to evaluate complicated cardiac malformation [10]. ECG-gated CT is utilized to diagnose heart diseases such as pulmonic stenosis, patent ductus arteriosus (PDA), and overriding aorta [11]. When TTE images are insufficient due to the heart’s complex morphology, ECG-gated CT gives us a better diagnostic insight.

This case report describes a case of UCSS where a definitive diagnosis using conventional echocardiographic examination was challenging; however, the defect was successfully identified using ECG-gated CT scans. This is the first reported case in which a dog was diagnosed with UCSS using an ECG-gated CT. It also documents transesophageal echocardiogram (TEE) as a valid supporting device for depicting the state of this condition.

## 2. History and Case Presentation

A 4-year-old castrated male Labrador Retriever was admitted to a primary veterinary clinic, where the dog initially presented signs of fatigue. During his check-up to determine the cause of his fatigue, he was originally diagnosed with ASD. However, since the defect could not be confirmed, he was referred to the Tokyo University of Agriculture and Technology Animal Medical Center to investigate further. Upon admission, there was no noticeable cardiac murmur during the physical exam, and his routine blood work was within the reference range, apart from the elevated value in aspartate aminotransferase (AST) and alanine aminotransferase (ALT), each being 186 U/L (reference value: 17–44 U/L) and 209 U/L (reference value: 17–78 U/L). Serum thyroxine (T4) level was 2.91 μg/dL, which was close to the upper limit of the reference range (1.00–2.90 μg/dL). N-terminal pro B-type natriuretic peptide (NT-proBNP) was elevated at 1200 pmol/L (reference value: ≤900 pmol/L).

The TTE was performed using LISSENDO 880LE (Fujifilm Ltd., Tokyo, Japan) equipped with the 9–12 MHz phase array transducer probe (Fujifilm Ltd., Tokyo, Japan). TTE revealed trivial tricuspid regurgitation (TR), a pulmonary to systemic blood flow ratio (Qp/Qs) of 1.28, and there were no signs of a flattened ventricular septum. A left-to-right shunt, indicated by an irregular blood flow from the vicinity of the LA to the RA, was seen, although based on the evidence given at that time, it was possible that the blood flow was not actually within the heart. Conclusively, ASD was strongly suspected, although there was no opening found at the typical location during the TTE (Figure 1). Otherwise, the TTE evaluation revealed that other parameters were within normal limits. The results from the TTE were insufficient for a definitive diagnosis, so plans were made to further investigate with an ECG-gated CT.

An ECG-gated CT was conducted using a 64-slice multislice CT scanner, Aquilion CXL (Canon Medical Systems Corp., Tochigi, Japan), on day 14 after initial consultation. At the time, the patient’s condition remained unchanged under no treatment. The dog was premedicated using butorphanol (Meiji Animal Health Co., Ltd., Kumamoto, Japan) (0.2 mg/kg, iv) and midazolam (Marubishi Pharm. Co., Osaka, Japan) (0.4 mg/kg, iv), induced with propofol (Nichi-Iko Pharm. Co., Ltd., Toyama, Japan) (6 mg/kg, iv), and maintained with isoflurane (Bussan Animal Health Co., Ltd., Osaka, Japan). No medication was administered to reduce cardiac frequency. As the contrast agent, Iopamidol (Hikari Pharm. Co., Ltd., Toyama, Japan) was administered intravenously through the right cephalic vein using an injector, A800 (Nemoto Kyorindo Corp., Tokyo, Japan), at 0.3 mL/s for 30 s with a total dose of 2 mg/kg. Bolus-tracking technique (SUREStart, Canon Medical Systems Corp., Tochigi, Japan) was used to manage time-to-peak enhancement; the scan was set to start automatically once the contrast agent reached the region of interest, the root of the aorta. For the starting scan, a threshold of 100 Hounsfield units was used for aortic opacification. The coronary CT angiography scan range was set to fit the entire heart, with a 15 mm margin at both the cranial and caudal ends. The parameters of helical retrospective ECG-gating were set automatically. The heart rate during scanning ranged from 64 to 75 beats per minute. For postprocessing, image datasets were reconstructed using AZE virtual place version 2.8.0.0 (Canon Medical Systems Corp., Tokyo, Japan), with a temporal window of 10% applied spanning from 0% to 90% of the R-R cycle. 

The ECG-gated CT showed communication between the LA and RA via the CS (Figure 2a). An enlargement of the CS, where the coronary artery measured 0.8 mm in diameter, whereas the CS measured 11.6 mm, was observed (Figure 2b). The CS drained normally into the RA; however, an ostium to the LA was seen, leading to the diagnosis of UCSS (Figure 3a). Under general anesthesia, TEE was also conducted using LISSENDO 880LE (Fujifilm Ltd., Tokyo, Japan) equipped with the 8–2 MHz phase array transesophageal probe (Fujifilm Ltd., Tokyo, Japan). With TEE, it was possible to identify the shunt as an abnormally large CS running along the cardiac wall (Figure 4).

There was no change observed in the dog’s condition and plans to keep observations were made. However, on day 130 of illness, an increase in the volume load in the right heart was suggested by the rise in right ventricular internal diameter end diastole (RVIDd) of 8.5 mm and right ventricular inflow of 69.3 cm/s, and trivial TR was observed. Along with exercise intolerance and worsening conditions, diuretics were prescribed (Furosemide, Nichi-Iko Pharm. Co., Ltd., Toyama, Japan) 0.5 mg/kg, po, BID). By day 157 of illness, the patient’s condition remained stable; however, the Qp/Qs ratio had decreased noticeably, and diuretics were removed. 15 months later, the condition remained stable, and we recommended that diuretics be prescribed again at their primary clinic if the patient experienced difficulty breathing.

## 3. Discussion

In humans, Shi et al. reported that out of 68,740 human patients treated for congenital heart disease, 159 cases were diagnosed with UCSS, resulting in the prevalence of 0.23%. In UCSS, persistent LSVC was commonly seen amongst them, and some of the common characteristics were cyanosis, dyspnea, palpitations, cardiac murmur, and arrhythmias [3,8]. The common complaints in human UCSS patients when arriving at the hospital were shortness of breath and dyspnea on excursion [12,13,14]. Although the patients exhibited similar symptoms, it has been suggested that CS anomalies may not produce significant clinical signs, and that UCSS cannot be diagnosed based on clinical signs alone [12,14].

In dogs, there are only three reported cases of UCSS, proving UCSS to be a rare condition in dogs as well [5,6,7]. The first case was reported by Zani et al. reported a case of a 5-month-old female French bulldog with severe pulmonary stenosis [7]. Although UCSS was not explicitly mentioned, a persistent left cranial vena cava that drained into the LA was found through left-atrial ventriculography. In addition, a bubble study verified the left-to-right shunt. These findings strongly suggest that there was an unroofed CS. This report highlights the importance of in-depth medical assessment, such as a CT exam, for diagnosis [5]. In 2023, Shin et al. reported a five-year-old castrated male Dachshund in which a UCSS was found due to cardiomegaly [6]. The patient had decreased activity and an enlargement of the right ventricle (RV). When auscultating the heart, a third heart sound at the left chest wall was heard. The blood test was within the reference range, but the value of NT-proBNP was elevated. When performing a cardiac catheterization, the catheter was entered into the LA through the defect in the CS, and UCSS was diagnosed. This patient was the first patient to both diagnosed and explicitly reported as a UCSS. Most recently, a case reported by Choi et al. described a 2-year-old spayed female Pomeranian where a UCSS was found asymptomatic [5]. There was no cardiac murmur detected, and blood tests were unremarkable as well. A subsequent CT confirmed a defect between the CS and the LA. Although the defect could be found through CT, performing an ECG-gated CT was recommended especially when considering surgical or interventional repair of the UCSS, for it provides a more precise image of the defect with less motion artifacts [5,15]. This was the first case of UCSS to be reported without symptoms.

Our patient’s activity was decreased, but no noticeable signs of an enlarged right heart were found based on TTE and thoracic radiograph at the time of initial consultation. As the condition progressed though, an enlargement and volume load of the RV was seen. No cardiac murmur was heard during the physical exam. The rise in AST and ALT was presumably due to hepatic congestion from volume load in the right heart. The elevated NT-proBNP level was likely due to the volume load in the right heart.

In the patient reported here, the TTE of UCSS showed similarities with ASD. Progressive RV enlargement, indicated by a gradual increase in RVIDd, was observed until day 157 after arrival. At first admission, there was no ventricular septum flattening, but after 130 days, a mild case of flat ventricular septum was noticed, indicating that the right heart had enlarged over time as well. Treatment of RV volume overload with diuretics resulted in an improvement in RV enlargement, as evidenced by a small decrease in RVIDd. The Qp/Qs ratio also increased until day 130 of illness, but after using diuretics, it became noticeably lower, showing that the outflowing blood volume from the RV to the pulmonary circulation had decreased due to the diuretics. As for the clinical symptoms, they did not progress during our follow-up.

When diagnosing UCSS, using just TTE may leave a risk of misdiagnosis [16]. In our case, the CS volume was overloaded. Considering that the mean diameter of the CS ostium in dogs is 5.5 ± 1.3 mm, the coronary sinus had become severely enlarged [17]. ECG-gated CT images showed that the diameter of the CS was 14.5 times larger than that of the coronary artery (Figure 2b). The largeness of the CS led to its appearance during the TTE as an irregular blood flow communicating between the LA to the RA, thus supporting the initial misdiagnosis of ASD. It is important to consider the possibility of extracardiac blood flow when no intracardiac defect is identifiable.

A correct diagnosis is brought with CT, although depicting the septal defect between the LA and the CS is not easy [6]. Evaluation of the heart using conventional CT, which is not synchronized with cardiac motion, is challenging because the images are often difficult to interpret. In human medicine, when related to coronary arteries, thoracic aortic ECG-gated CT angiography, able to create a clear image of the heart chamber owing to its characteristic of operating simultaneously to an ECG, is now regarded as the method of choice [18]. By synchronizing with an ECG, a CT image of one heartbeat can be created, providing an accurate image of the diastolic and systolic cardiac morphology. 

A great supporting tool for the diagnosis of UCSS is TEE. In our patient, the blood flow from the CS to the RA was better visualized on the TEE than on the TTE. Shi et al. reported that out of all 159 human patients who had TEE performed, UCSS was confirmed in 91 cases (57.2%) [19]. TEE is widely recognized for its utility in diagnosing congenital heart diseases, including PDA [20]. 

When a severe case of UCSS is discovered in humans, the standard approach for treatment is a surgical repair. Surgical repair of UCSS is performed under general anesthesia and hypothermic cardiopulmonary bypass. Techniques include closure with an atrial septal patch, reroofing of the CS using pericardium, or atrial reconstruction in cases where LSVC is not involved [19]. In the veterinary field, although operating under hypothermic cardiopulmonary bypass is technically possible, it is often unfeasible in current clinical standards. It is noteworthy that in our case, a mild case of UCSS was managed effectively using diuretics, without a surgical repair.

## 4. Conclusions

In this patient, conventional echocardiography showed no direct evidence towards a definitive diagnosis. For a condition like USCC, which is difficult to diagnose non-invasively, a correct diagnosis was only made after an ECG-gated CT examination. On an ECG-gated CT, blood vessels are clearly depicted, allowing us to see the blood flow from the LA to the CS in detail. Furthermore, despite its rarity, UCSS should be considered when the CS appears enlarged [6]. In the case of UCSS, where a simple TTE is insufficient for diagnosis, conducting an ECG-gated CT promptly proves to be crucial.

## Figures and Tables

**Figure 1 animals-15-02834-f001:**
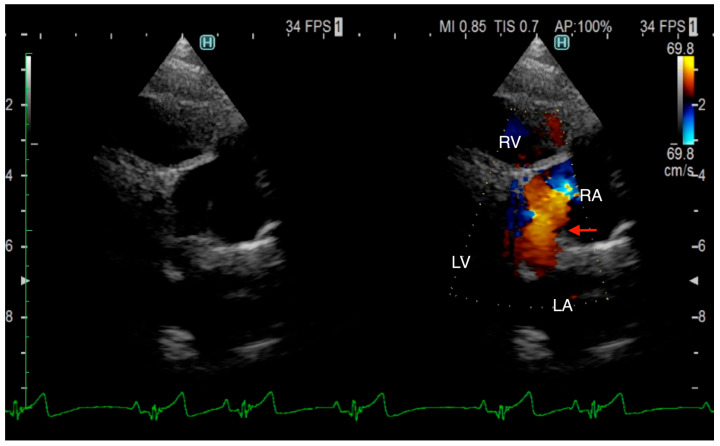
Right parasternal long-axis four-chamber view of transthoracic echocardiography at the time of arrival. Transthoracic echocardiography shows an irregular communication between the left atrium and right atrium (arrow). From this image, it is unclear whether the blood flow is within the heart. RA, right atrium; RV, right ventricle; LA, left atrium; LV, left ventricle.

**Figure 2 animals-15-02834-f002:**
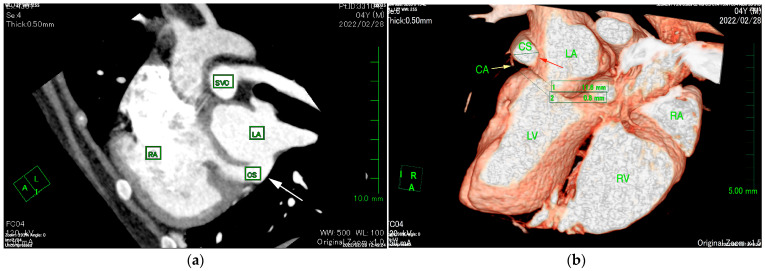
(**a**) ECG-gated CT multiplanar reconstruction (left oblique sagittal view) demonstrating the irregular communication between the RA and LA (arrow) that is formed by the unroofed CS. The opening in the CS allows blood flow from the RA to the LA. (**b**) ECG-gated CT volume rendering (sagittal slice, left lateral view, observed from the right) showing the markedly enlarged CS (red arrow) compared to the CA (yellow arrow). The CS measures 11.6 mm in diameter and is abnormally large compared to the CA, measuring 0.8 mm. CS, coronary sinus; CA, coronary artery LA, left atrium; LV, left ventricle; RA, right atrium; RV, right ventricle; SVC, superior vena cava.

**Figure 3 animals-15-02834-f003:**
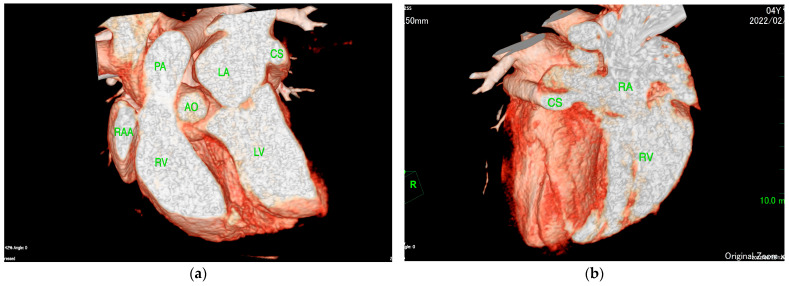
(**a**) ECG-gated CT volume rendering (sagittal slice, right lateral view, observed from the left) showing the CS connecting to the LA. (**b**) ECG-gated CT volume rendering (oblique sagittal slice, right lateral view, observed from the right) showing the CS connecting to the RA. CS, coronary sinus; RA, right artery; RV, right ventricle; LA, left atrium; LV, left ventricle; AO, aorta; PA, pulmonary artery; RAA, right atrial appendage.

**Figure 4 animals-15-02834-f004:**
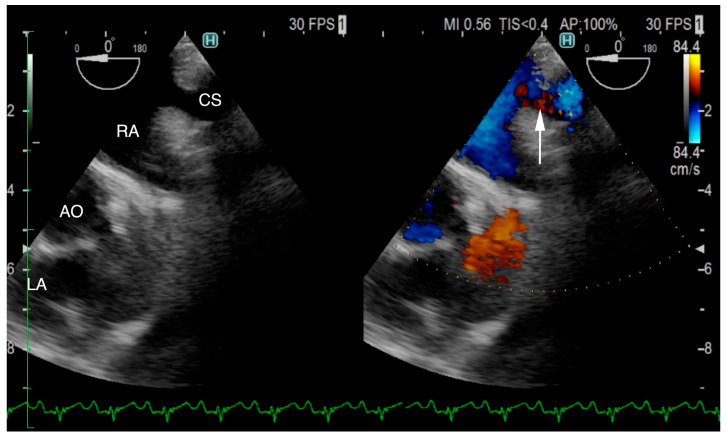
Transesophageal echocardiography (TEE) shows the coronary sinus (CS) draining into the right atrium (RA) (arrow). The CS is typically not visible on TEE; however, in our patient, it is seen, suggesting that it is dilated due to an unroofed segment or an opening in the CS. CS, coronary sinus; RA, right atrium; AO, aorta; LA, left atrium.

## Data Availability

The data presented in this report are available on request.

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
