# Peer review of "Unroofed Coronary Sinus in a Dog: Diagnostic Utility of ECG-Gated Computed Tomography"

_animals, 2025, doi:10.3390/ani15192834_

Round 1

Reviewer 1 Report

Comments and Suggestions for Authors

Thank you very much for sharing this very interesting research report.

This case report describes the successful definitive diagnosis of canine UCSS using ECG-gated CT, despite the difficulty in diagnosis via ultrasound. It is a highly interesting report with two important novelties: the diagnosis of USSS, which is extremely rare in dogs, using ECG-gated CT.

However, further information is needed to justify the necessity of ECG-gated CT.

The following are specific comments.

Major Comments

  1. The authors state that ECG-gated CT was useful for diagnosing UCSS, but provide no information regarding the CT protocol. Specifically, no information is provided about the equipment used (information about the ultrasound equipment is provided). If CT examination is recommended, the reviewer hopes the authors will explicitly state the CT protocol used by the authors. ECG-gated CT is not yet a common examination in veterinary medicine, and information is lacking. If the authors present their protocol, readers could perform ECG-gated CT based on this report. This information will further strengthen the value of this case report. Particularly, details such as the image acquisition method and heart rate during scanning are crucial information for ECG-gated CT.
  2. Recently, similar reports focusing on genetic findings have been published. Please explain the differences from this report. If citation is necessary, the reviewer recommend citing them.

Choi Y et al., Case Report: A rare case of asymptomatic unroofed coronary sinus in a dog: diagnostic imaging and genetic findings. Front Vet Sci. 2025;12:1611021.

Minor Comments

Reports on ECG-gated CT in dogs are limited, but there have been reports on its diagnostic utility for PDA, among other things.

To indicate that this report is not a “clinical study” but rather an ECG-gated CT examination for evidence-based diagnosis, it might be better to include past reports in dogs in the background section.

Lines 53-56

Unlike in humans, is this the only reported case of UCSS in animals? Is this specifically regarding dogs? If so, it would be clearer to explicitly state “in dogs.”

Lines 54-56

Is the placement of this sentence appropriate? Would it be more suitable in the final paragraph?

Line 81 「but due to its uncertainty」

Does this mean the defect could not be confirmed?

Line 85

This correction is left to the editors and proofreaders, but shouldn't the notation be “186 U/L” rather than “186U/L”? Numerous other instances can be confirmed.

Lines 93-96 and Figure 1.

 The authors state, “Transthoracic echocardiography shows an irregular communication between the left atrium and right atrium (arrow), caused by the unroofed coronary sinus.” Does this mean UCSS was diagnosed via ultrasound?

Figure 2

Please explain the difference between the red and white arrows.

Figure 2, 3, 4

The images are very difficult to recognize. White text on a white background is hard to discern. Please modify the layout as well.

Lines 191-192

Based solely on this report, can we generalize that mild UCSS can be controlled with diuretics?

Lines 198-200

You compare CS size to coronary artery size, but what is the normal range? The reviewer also agrees on CS enlargement, but please clarify.

Lines 214-216

 retrospective ECG-gated CT angiography? Prospective electrocardiographic triggering reduces radiation exposure because it images only the necessary cardiac phase.

Reviewer 2 Report

Comments and Suggestions for Authors

I find that this manuscript provides valuable insight into a very rare congenital heart disease in dogs—Unroofed Coronary Sinus Syndrome —and demonstrates the diagnostic utility of ECG-gated CT, which may improve accuracy in clinical practice. However, there are some points that require revision. 

Introduction)

Line 53: It seems that what you intended to convey is that only one dog case has been reported. Please clarify that it is a dog, and confirm whether it is indeed just one case, as the Discussion mentions two cases. Also, mentioning that Shin’s case involved surgery seems sufficient in the Discussion.

Lines 54–56 appear again at the end of the Introduction, so they should be deleted.

Lines 71–73 would be more appropriately merged with the second paragraph of the Introduction.

Case presentation)

Line 80: Were fatigue symptoms the only clinical signs? If so, on what basis was ASD diagnosed at the primary clinic?

Line 86: A T4 value of 2.91 exceeds the reference range you provided. Therefore, stating “within normal range” is incorrect.

Table 1: What does the TR “+” on day 0 indicate? Is Table 1 really necessary, or would it be better moved as a supplementary table?

Line 119: Does “day 14 of illness” refer to 14 days after the onset of fatigue? In the table above, the dates are based on the initial consultation, which is confusing. Please unify the date references. Also, were there no other treatments administered before day 14? Was the patient’s condition similar during that period?

Figures 2, 3, 4: It might be better to combine these into two figures. Additionally, except for Figure 2, the labels on the images (LA, LV, RA, RV, etc.) are not clearly visible; please adjust the label color. For Figure 4a, reduce the margin and enlarge the heart for better visualization.

Line 148: What does “stable” mean? Did the initial symptom of fatigue resolve? Please revise to use expressions more appropriate for a scientific article.

Line 150: The RVIDd value differs from what is presented in Table 1.

Discussion)

Line 165: Since UCSS is a congenital disease, why is it associated with middle- to senior-aged patients in humans? How is this relevant to your case?

Line 167: Apart from Shin’s case, are there no other canine cases for comparison? Please discuss similarities and differences between your case and other reported cases, citing the relevant literature. Additionally, why were Shin’s case and other canine cases able to be diagnosed without ECG-gated CT? This should also be addressed.

Line 169: What does the reference “(Shin)” indicate?

Line 173: The expression “Shin’s patient” may not be appropriate; please rephrase. Also, make it clear in this paragraph that the discussion refers to your patient.

Line 177: This sentence appears unnecessary.

Figure 6: Its appearance in the Discussion section seems inappropriate. Consider moving it to the Case Description section or deleting it.

Line 185: The sentence beginning with “During the ~” appears unnecessary.

Line 186: The statement that RV volume overload improved with diuretics alone in your patient is noteworthy; however, it may be more appropriate to present this in the context of the final paragraph of the Discussion. For example, in humans, surgery is a treatment option for severe cases, which is not easily feasible in veterinary practice; only one canine case has been reported, whereas your patient was successfully managed with diuretics alone.

Line 222: If surgery is one option for severe human cases, what are the alternative treatment options? How are non-severe cases managed?

Conclusion)

Line 230: This sentence appears unnecessary.

Lines 230–232: What is meant by “conditions like UCSS”? Is it really the case that a diagnosis can only be made after an ECG-gated CT?

Lines 233–236: The content here is repetitive. Since your manuscript is a single case report, please present the significance of your study in a concise and clear manner.

Comments on the Quality of English Language

The English language is generally understandable; however, from the perspective of scientific writing, improvements are needed in clarity, consistency, and the use of professional terminology.

Round 2

Reviewer 1 Report

Comments and Suggestions for Authors

Thank you for the revised manuscript. The authors have appropriately responded to the reviewers' questions.

Several points in the revised manuscript require further minor revision. If these points are appropriately revised, the reviewer will not provide further comments.

The following are specific comments.

Minor Comments

Line 123

Please also note that it is a “64-slice multislice CT scanner”.

Line 128 intravenously

Right cephalic vein?

Line 129 Bolus-tracking technique

 If the bolus tracking technique was used, please also note the administration time.

Line 131 scan delay

 What is the scan delay set to in seconds?

Line 132 aorta

 Ascending aorta?

Figure 2, 3

 Please add the cross-sectional names of MPR and VR to the Legends.

Author Response

Please see pdf file.

Reviewer 2 Report

Comments and Suggestions for Authors

I have carefully reviewed your response letter and the revised manuscript. Overall, the revisions have been well addressed; however, a few additional modifications are still required.

In particular, for the next revision, please ensure that the response letter includes accurate line numbers.

Line 52: Please change “few cases” to “3 cases”.

line 56: The phrase “To date~” should be deleted. In addition, citations for the three cases are required.

Line 92: it would be more accurate to revise the sentence as follows:
“The serum thyroxine (T4) level was 2.91 μg/dl, which was close to the upper limit of the reference range (1.00–2.90 μg/dl).”
Referring to the value as 2.9 rather than 2.91 could be considered misleading.

Page 5: It appears that Figures 2 and 3 in the revised manuscript have been switched. This is evident when comparing the placement in the main text, the figure legends, and the previous version of the manuscript.

Line 176: [9,10,11]

Line 180: Since your case is not a previously reported case, this sentence should not be written in that way.

Lines 180–198: This paragraph reads as a list of descriptions, making it difficult to clearly identify which case is being discussed and what your main point is. Please revise more concisely, including only the necessary explanations.

Line 248: This sentence is grammatically awkward. Please revise it to something like: “For a condition like USCC, which is difficult to diagnose non-invasively, a correct diagnosis was only made after an ECG-gated CT examination.”
or “A correct diagnosis of a condition such as USCC, which is difficult to diagnose non-invasively, was only achieved after an ECG-gated CT examination.”

Comments on the Quality of English Language

Although the English has improved compared with the previous version, the manuscript would still benefit from additional language polishing to enhance clarity, consistency, and readability.

Author Response

Please see pdf file.
